# Effectiveness of an E-Book App on the Knowledge, Attitudes and Confidence of Nurses to Prevent and Care for Pressure Injury

**DOI:** 10.3390/ijerph192315826

**Published:** 2022-11-28

**Authors:** Shu-Ting Chuang, Pei-Lin Liao, Shu-Fen Lo, Ya-Ting Chang, Hsiang-Tzu Hsu

**Affiliations:** 1Taichung Tzu Chi Hospital, No. 88, Section 1, Fengxing Road, Taichung 427, Taiwan; 2Department of Nursing, Tzu Chi University of Science and Technology, No. 880, Section 2, Jianguo Road, Hualien 970046, Taiwan; 3Tzu Chi Fondation Tzu Chi Taichung Home-Based Long-Term Care Facilities, No. 88, Section 1, Fengxing Road, Tanzih District, Taichung 427, Taiwan; 4Department of Nursing, Tzu Chi University, No. 701, Section 3, Zhongyang Road, Hualien 97004, Taiwan; 5Tzu Chi Foundation Long-Term Care Promotion Center, No. 88, Section 1, Fengxing Road, Taichung 427, Taiwan

**Keywords:** E-book app, pressure injury, knowledge, attitudes, confidence, nurses, nursing

## Abstract

Aims: This study evaluates the effectiveness of an interactive E-book app training program in improving nurses’ knowledge, attitudes, and confidence to prevent and care for pressure injury. Design: Randomized experimental study. Methods: Participants were recruited from a teaching hospital in Taiwan. The study was carried out between 20 March 2014 to 1 April 2016. In total, 164 participants were randomly assigned to a pressure injury E-book app training program (n = 86) or a conventional education program (n = 78) with a one-month follow-up. Outcome variables were levels of pressure injury knowledge, attitudes, and confidence of pressure injury care. Results: Participants answered 51.96% of the pressure injury knowledge questions correctly before the intervention and 75.5% after the intervention. The pressure injury attitude score was slightly positive, with moderate confidence in pressure injury care. The knowledge, attitudes, and confidence of pressure injury care of the two groups in the pretest and posttest groups increased significantly. Analysis of covariance indicated that nurses in the pressure injury E-book app group had significantly greater improvement in knowledge, attitudes, and pressure injury care confidence as compared with the control group. Conclusion: The pressure injury E-book app interactive training program was effective in improving nurses’ knowledge and attitudes toward pressure injury care and in enhancing their confidence in pressure injury care; therefore, this program has potential for nurses’ in-service education in both Taiwan and worldwide. Impact: E-book apps allow individuals to control the time and place of learning. Direct observation of procedural skills can provide feedback to trainees on techniques to ensure learning effectiveness and pressure injury care quality.

## 1. Introduction

With the changes in global population structure, the aging population, and the increase in comorbidities, pressure injuries (PIs) have become an important public health issue [1]. In particular, hospital-acquired PIs (HAPIs) can be prevented and are currently an important indicator of the quality of care [2,3]. However, in their meta-analysis of 39 articles, Li et al. [3] highlighted that the PI prevalence rate was 12.8% and the incidence rate was 8.4% among hospitalized adults. Medical device-related PIs comprise 12% of all PIs [2]. The most frequently occurring stages were reported to be Stage I and Stage II. Severe PI has contributed to prolonged hospital stay, an increase in life-threatening consequences, huge medical expenses, and even death [4]. In the United States, HAPIs were reported to be the cause of death of 60,000 people, with an average cost of $10,708. Moreover, Stage 3 and Stage 4 PIs account for 58% of all HAPI costs [5]. Inadequate training is an important component of unsafe care. The main vision of the Global Patient Safety Action Plan 2021–2030 was to prevent harm in health care and to ensure that each patient receives safe and respectful care, every time, everywhere. One of the important strategies of this plan is to provide “health worker education, skills and safety” [6]. Nursing education that lacks systematic and structured wound courses and sufficient time to learn systematic knowledge of PI care results in insufficient knowledge, negative attitudes, and lack of confidence in wound care [4,7]. Many different approaches to wound care training have enhanced nurses’ knowledge and clinical competency to prevent the development of PIs and to promote wound healing [1]. However, the current shortage of nurses and high workload of clinical workers hinders these professionals from participating in face-to-face education courses; thus, there is a need to establish an innovative PI training program [8,9]. For this reason, the development of innovative, interactive, and empirical training strategies is currently an important clinical topic.

### Background

Smartphone use began increasing globally in 2011, and since then, E-book readers have become widely used [10]. The development of mobile and information technology has changed teaching methods and the way information is transmitted. Interactive learning materials are becoming widely common and available. As a result of the rapid, increased use of smartphones, there has been rapid development of E-book apps, which are effective as teaching methods that not only can provide nurses with evidence-based knowledge but also improve clinical care skills in traditional clinical practice. O’Connor and Andrews [11] evaluated 200 undergraduate nursing students in the United Kingdom in terms of the effect of smartphones and mobile apps on clinical learning and reported that smartphone apps help students learn in clinical practice. Nason et al. [10] conducted a survey of 36 urology interns in Ireland regarding the use of a smartphone E-book app in clinical care. Most of the trainees downloaded medical or urology apps, and 86.8% of users thought it was of great help to their clinical work. Nursing staff often use smartphones extensively to search for care information, but they hope to provide systematic, evidence-based information and make care decisions in the clinic [12]. The multifunctional interface of the smartphone E-book app allows users to systematically learn medical care information, access multimedia related to care, and practice skills, with the result of reducing medical errors and enabling more efficient decision-making [10]. At present, several studies have been conducted using E-book interventions to improve learners’ knowledge, motivation, and competence as Appendix A for nursing education and internship guidance. However, the subjects of this research have mostly been nursing students [13]. Sung and Park [13] used a mobile app-based cultural competence training program with 49 nurses from South Korea and found that the cultural competence of participants can be improved. However, that study used only one group with a pre- and posttest intervention design, which weakened the strength of the research inferences. PI is recognized as an important indicator of patient safety and quality of care in hospital settings. However, there is a lack of empirical research on the use of mobile devices, such as smartphones, among nursing staff members enrolled in in-service education training. Thus, the purpose of this study was to compare the education strategies of an interactive E-book learning program and a conventional education program to assess the impact on PI care-related knowledge, attitudes, and confidence of nurses working in a hospital.

## 2. The Study

### 2.1. Aims

The purpose of this study was to determine whether a PI E-book app would improve nurses’ PI-related knowledge, attitudes, and care confidence.

### 2.2. Research Hypothesis

The study had two hypotheses: (1) Nurses in the experimental group will have a higher score of PI care knowledge, attitudes, and confidence as compared with nurses in the control group after undergoing a PI E-book app training program. (2) The learning effect will change with follow-up time (i.e., there will be an interaction effect between experimental group and follow-up time).

### 2.3. Design

This was a randomized, controlled, unblinded clinical trial with two groups. Nurses were randomly assigned to the intervention group or a control group by computer-generated allocation. The author, Lo, S.F, confirmed the random number generation by a clinical trial statistics expert; then, participants were assigned and enrolled to interventions by the author Liao, P.L. The intervention group attended the PI E-book app training program, and the control group received a traditional lecture program with similar content. The first week comprised the pretest phase, during which participants completed the pretest questionnaire. In addition, the two groups attended traditional, 60-min, face-to-face instruction lectures with PI PowerPoint slides conducted by an internationally certified wound-care therapist.

This trial was carefully designed to conform to the CONSORT statement The two groups were evaluated after the one-month intervention. The author, Chang, Y.T., conducted measurement outcomes.

### 2.4. Ethical Consideration

This study was reviewed and approved by the Taichuug Tzu Chi Hospital Research Ethics Review Board (Institutional Review Board No. REC104–38) in Taiwan. Based on ethical considerations, the research team personally explained the purpose and methods to the nurses at the nursing meeting. Nurses were informed that they could withdraw from the research process at any time without effects on their work performance appraisal rights. Signed informed consent was obtained from participants after they had been informed about the contents of the study.

### 2.5. Participants

The participants were a convenience sample of all registered nurses (RNs) from a 1000-bed teaching hospital in Taiwan. Study inclusion criteria were as follows: (1) aged 20 years or older, (2) working in a ward that cares for PI patients, and (3) working in the ward for longer than three months. The exclusion criteria were (1) part-time RNs and (2) wound/stoma specialist nurses, head nurses, or nurse practitioners.

### 2.6. Power Calculation

The sample number was calculated by power analysis using G*Power version 3.1. We used the confidence of PI care scale as the primary outcome indicator. The secondary outcome measurement was PI-related knowledge scales and attitudes scales. We then computed the required sample size based on two independent groups, two-tailed, with an effect size of 0.5, alpha set to 0.05, and a power of 0.8 for estimation. A sample size of 64 was required for each of the two groups. To maintain the attrition rate within 20%, when the power was 0.8, the number of samples in each group increased from 64 by 20%, and the number of cases was at least 77.

### 2.7. Interventions

#### 2.7.1. Experimental Group

The development of the interactive PI E-book app training program was based on the 2014 guidelines on the prevention and treatment of pressure ulcers [14]. SimMAGIC software (Hamastar Technology Company, Kaohsiung, Taiwan) was used to edit the interactive PI E-book. To improve the overall multimedia effect, we integrated pictures, audio, videos, film, and animations into the E-book. At the same time, the E-book included functions such as notes, bookmarks, and preset question prompts to improve the learning effect of the subject [15]. The PI E-book app (Figure A1) comprised of three components: the first main section contained information on the definition of PI, etiology, staging; the second part included risk factor assessment, preventive skin maintenance, dressing selection, nutrition support, repositioning, and wound care; and the third comprised of interactive learning feedback and practice with participant self-learning in every session. When the participants were assigned to the experimental group, the author, PL, taught them how to download the mobile app and how to use the E-book. Participants in the intervention group downloaded the PI E-book app for their iOS or Android smartphone and also attended lecture teaching. Each PI E-book app session took 10–15 min on average. During the study period, the experimental group could read PI E-book app content anytime, anywhere.

#### 2.7.2. Control Group

The control group received traditional small-group classroom teaching, and the content was the same as given in the PI E-book app program in the experimental group. On average, the lecture program took 60 min. 

### 2.8. Data Collection

In this study, a demographic data sheet was used to record the participant data. The study was carried out between 20 March 2014 to 01 April 2016. The PI knowledge scale, PI care attitude scale, and PI care confidence scale for these self-reported measures were collected before and after the intervention. The four instruments are described below.

#### 2.8.1. Demographic Form

A demographic form was used to collect the subjects’ demographic characteristics, which included age, sex, marital status, level of education, nursing level, in-service PI training perception, previous training, and so forth.

#### 2.8.2. Knowledge of Pressure Injury Scale (KPIS)

The KPIS was designed to measure nurses’ knowledge of PI care based on the 2014 PI guidelines [14] and related wound bed preparation literature [16]. It consisted of 36 items including PI risk factors, assessment, signs and symptoms of infections, dressing choice, skin protection, prevention of PI, and wound care. To avoid having the participants guess when filling out the answers and to improve the credibility of the responses, we adopted a single-choice question type. Each item included four options. After the scales were collected, the scores were converted. The correct answer was assigned 1 point. The wrong answer was given 0 points. The KPIS had a range of possible scores from 0 to 36. The higher the score, the better the understanding of PI.

#### 2.8.3. Attitude of Pressure Injury Scale (APIS)

The APIS scale was designed to measure nurses’ attitudes regarding PI prevention and wound acre based on PI attitude-related references [17]. The APIS scale comprised of 27 items rated on a five-point Likert-type scale (1 = strongly disagree, 5 = strongly agree). Scores on the negatively worded items were reversed and calculated to obtain the total attitude score. The APIS had a range of possible scores from 27 to 135. A higher score indicated more positive attitudes for caring for PI patients.

#### 2.8.4. Confidence of PI Care Scale (CPICS)

The CPICS was designed to understand the level of confidence of the participants in PI assessment, prevention, and wound care [17]. This scale includes 33 items that measure nurses’ confidence in PI care. Respondents answered each of the statements with a five-point Likert-type scale, ranging from one to five (1 = not at all, 5 = extremely), which ranged from 33 to 165, with higher scores corresponding to more positive outcome intentions related to confidence in PI care.

### 2.9. Validity and Reliability

To verify the validity of the questionnaire, we invited five internationally certified enterostomal therapists to examine and assess the suitability of the instrument. The scales were pilot tested with 30 on-the-job nursing students for face validity. In addition, Cronbach’s α was used to test the reliability of the instrument. The KPIS internal consistency reliability (Cronbach’s α) was 0.91. A Cronbach’s α coefficient of 0.81 was reported for APIS. A Cronbach’s α coefficient of 0.90 was reported for CPICS.

### 2.10. Data Analysis

The SPSS software, version 20.0 (SPSS Inc., Chicago, IL, USA) was used for descriptive and inferential statistics, with a *P* value < 0.05 being considered significant. Descriptive statistics use frequencies, percentages, averages, and standard deviations. The homogeneity test of the subjects’ characteristics between the control group and the intervention group was based on the chi-square test or Fisher’s exact test and two independent samples’ *t* tests. We used the paired *t* test or Wilcoxon test to determine the differences between the intervention and control group on gain scores and baseline scores of the KPIS, APIS, and CPIS. An analysis of covariance (ANCOVA) regression was used to determine the posttest difference between the intervention and control groups.

## 3. Results

In this study, a total of 180 nurses were screened for research qualifications. Of those, four (1%) were found to be ineligible for the following reasons: (1) did not meet inclusion criteria or (2) refused to join this study. The remaining 176 individuals who met the study eligibility criteria agreed to participate and were randomized; 12 nurses (2 experimental, 10 control) were excluded from analyses because their schedule prevented them from completing this study, and the dropout rate was 6.8%. Eighty-six nurses in the experimental group and 78 nurses in the control group were included in the data analysis (Figure 1).

### 3.1. Demographic Characteristics

The experimental and control groups were compared at baseline (Table 1). Participant ages ranged from 22.63 to 54.10 years, with a mean of 31.28 years (SD = 6.29). Clinical working experience ranged from 0.5 to 28.17 years, with a mean of 6.03 years (SD = 5.52). Most participants were unmarried with a college education level (63.4%) and N2 level in nursing grade (43.3%). Although 72.6% of the participants received PI-related courses in school, nearly half believed it was not enough. In-service PI education and training did not meet their needs for clinical PI care. Of the participants, 68.3% did not attend PI-related courses, and most (75%) had not read PI literature. However, we found a significant between-group difference in receipt of in-service PI training courses (*t* = 5.11, *p* = 0.024).

### 3.2. Effects of the Intervention

#### 3.2.1. Comparison of PI Knowledge

The mean pretest score for perceived KPIS for the subjects in this study was 14.55 (SD = 3.62). Individually, the rate of correct answers for PI knowledge was only 40.41% in the pretest (Table 2). To examine the effect of the PI E-book app, we used a paired *t* test to examine the change in KPIS from pretest to posttest in each group. Statistically significant differences were demonstrated the experimental group (*t* = −23.89, *p* < 0.000, 95% confidence interval [CI]: −9.98 to −8.54) and control group (*t* = −8.98, *p* = 0.010, 95% CI: −2.80 to −1.78). Table 3 presents the pretest and posttest scores for KPIS for the experimental and control groups.

We used ANCOVA to analyze the effectiveness of the PI E-book app. Receipt of an in-service PI education course was significant, and the pretest scores for the KPIS were entered as a covariate. The assumption of homogeneity of the regression slope (*F* = 7.58, *p* = 0.058) was not violated in the ANCOVA. Excluding the influence of receiving an in-service PI education course and baseline KPIS, there was a significant difference in the KPIS between the two groups (*F* = 98.94, *p* < 0.000).

#### 3.2.2. Comparison of APIS

The mean pretest score for perceived APIS was 84.55 (SD = 12.37). The PI attitude score of the participants was slightly positive, but we found a large difference in the pretest score range (Table 2). We used a paired *t* test to examine the changes in APIS scores from pretest to posttest for the two groups (Table 3). Statistically significant differences were demonstrated in the experimental group (*t* = –24.73, *p* < 0.000, 95% CI: −23.95 to −20.39) and the control group (*t* = –10.24, *p* <0.000, 95% CI: –10.10 to –6.81). We used receipt of in-service PI education course and baseline APIS as covariates in the ANCOVA analysis. The assumption of homogeneity of regression slope (*F* = 0.52, *p* = 0.470) was not violated for the ANCOVA. Excluding the influences of in-service PI education course and baseline APIS, we found a significant difference in the APIS between the two groups (*F* = 33.20, *p* < 0.000).

#### 3.2.3. Comparison of CPCIS

The mean pretest score for perceived CPCIS for the subjects in this study was 106.57 (SD = 20.84). Participants had a moderate level of confidence in PI care in the pretest (Table 2). To examine the effect of the PI E-book app, we used a paired *t* test to examine the change in CPCIS from pretest to posttest in each group. Statistically significant differences were demonstrated the experimental group (*t* = −20.51, *p* < 0.000, 95% CI: −30.20 to −24.86) and control group (*t* = −12.52, *p* < 0.000, 95% CI: −15.00 to −10.89). Using the in-service PI education course and baseline scores for confidence as covariates in the ANCOVA analysis, the assumption of the homogeneity of the regression slope (*F* = 4.46, *p* = 0.067) was met for the analysis. Excluding the influences of the in-service PI education course in the two groups demonstrated a significant difference (*F* = 32.71, *p* < 0.000).

## 4. Discussion

Nurses have an important responsibility for preventing PI and promoting wound-healing in clinical care facilities. A key factor in PI prevention and management is a nursing staff with sufficient knowledge, positive attitudes, and enough confidence and competency in PI care [18]. The PI E-book app training program successfully increased the nurses’ knowledge of PI care and positive attitudes toward evidence-based PI management. Increases in PI care confidence were evident at both 1 and 4 weeks after the PI E-book app intervention. The findings highlight the effectiveness of an innovation technology learning program not only for improving knowledge, attitudes, and care confidence but also for maintaining changes over time.

### 4.1. Influence of the PI E-Book App on Nurses’ Knowledge

The biggest challenge in current clinical wound care comes from the lack of adequate knowledge of wound care among nurses and the poor links between evidence and wound care in practice [4,17]. Therefore, it is necessary to construct systematic and structured wound care training programs in both nurse education and continuing education after graduation [18]. Insufficient knowledge of wound care has a negative effect on the effectiveness of nurses’ clinical care [4,18]. Equipping nurses with the relevant specialty knowledge and skills is crucial to enable positive effects. Training programs should include wound assessment, dressing choice, and wound care [19]. Although numerous studies on nurses’ knowledge of a PI training program revealed contradictory findings [4,9,17], the consensus is that a systematic curriculum based on evidence is very important for improving PI knowledge [4,8,9]. In particular, repetitive learning can strengthen the construction of clinical knowledge [9]. The use of an E-book app can provide learning at any time to help nurses construct PI knowledge [15]. In this study, although the experimental and control groups had low knowledge scores on the pretest, the two groups improved their PI knowledge, whether they used e-books or traditional teaching methods. Like Western countries, most nursing schools in Taiwan only teach simple wound and drainage tube care in basic nursing experiential courses, which limits their knowledge of wound care [20]. In this study, participants in both the experimental and control groups increased their knowledge after the systematic course. Moreover, 75% of the nurses in this study had not read any PI articles in the past year. On-the-job education can indeed improve nurses’ knowledge. In particular, the E-book app intervention significantly improved the nurses’ knowledge levels of PI prevention and wound care [21]. Therefore, this study demonstrates that an E-book app might play an important role in improving the knowledge of nurses who care for patients with PIs.

### 4.2. Influence of the PI E-Book App on Nurses’ Attitudes

The attitude of the nurse is an important factor that influences nursing confidence and competency [17,18]. A positive attitude will trigger the implementation of PI preventive strategies and spur the nurse to take appropriate care actions based on the wound condition to promote wound healing [17]. Beeckman et al. [17] conducted a survey of 553 nurses regarding their knowledge and attitudes toward PI care. They found that nurses’ attitudes toward PIs are significantly correlated with the application of adequate prevention. These findings are comparable to those of Karimian et al. [22], who found that the attitude of nurses caring for PI patients can be improved through educational videos. Beeckman et al. [23] also found that nurses’ attitudes were significantly improved when they had recently joined a multifaceted, tailored implementation intervention on PI. In this study, the experimental and control groups had negative attitudes before the intervention. This might be related to the fact that nearly 70% did not receive an in-service PI education course before participating in the study. A large proportion of the participants pointed out that school wound care courses and in-service PI training are insufficient. The E-book app significantly improved the nurse’s attitudes toward PI prevention and wound care. Therefore, this study demonstrates that a systematic and structured education program can improve attitudes toward PI care.

### 4.3. Influence of the PI E-Book App on Nurses’ Confidence

The PI training program must combine theory, evidence, and practice, providing the information needed to make well-informed clinical decisions to maintain nurses’ clinical confidence and ability [1,24]. In particular, training content should increase PI clinical judgment skills [1]. Previous research results indicate that both team-based learning [25] and lecture-based learning [26] for nursing staff members can improve PI care behavior and confidence scores. Sung and Wu [24] indicated that nursing students learning a community health nursing course through an E-book increased their cognitive skills and problem-solving ability as compared with traditional teaching. Indeed, Lin et al. [27] identified that nurses who accept multimedia and an interactive E-book teaching approach can enhance their knowledge of arrhythmia and improve their related practical skills. In this study, the experimental and control groups had low confidence scores on the pretest, but the E-book app intervention significantly improved the nurses’ confidence levels in PI prevention and wound care. If nurses had enough wound care knowledge in etiology, assessment, and management, their confidence and competency would increase and help improve quality of care and patient safety [28]. This study presents an important new finding, that the E-book app approach is effective in improving nurses’ care of PI.

### 4.4. Study Limitations

This study has a number of limitations that provide a focus for future research. First, we used participant self-assessment of interventional effectiveness, and we recommend that future studies use direct observation of procedural skills to evaluate and provide feedback to trainees on clinical operation techniques to ensure learning effectiveness and quality of PI care. Second, the participants in this study were from only one hospital. We suggest that future studies should explore longitudinal analyses among health care settings and long-term care facilities. Third, study participants were revisited for only four weeks. An interactive learning app allows learners to watch and learn at any time, and future studies with 1-, 6-, and 12-month follow-up periods might prove informative [9]. Four, the PI training program was mainly used to reduce the incidence of PIs. Future research should explore whether the incidence of PIs is reduced after an education intervention to respond to the goal of patient safety care. This study was performed between 2014 to 2016, which leads to older research results. However, according to Martinengo et al. [21], who carried out a systematic review of seven wound education studies for nursing students and clinical nurses, the results show that the intervention majority was online digital education, not interactive learning app design. Smartphones are popular and functionally enhanced, and future research can verify the effectiveness based on these study results.

## 5. Conclusions

The nurses shortage is a growing and urgent concern worldwide. How nursing staff can improve their PI knowledge and enhance the quality of care through effective learning methods without affecting their personal time is currently an important issue in clinical nursing [29]. E-book apps allow them to self-regulate their learning, manage their time, seek assistance, and perform self-evaluations [30]. Innovative education strategies offer a promising alternative for PI prevention and care for nurses. The E-book app has the potential to not only effectively improve nurses’ knowledge of PI and strengthen their confidence in wound management but improve their ability to perform safe and reliable professional wound care activities [22]. This report is the first empirical study on the effect of an E-book app on nurses’ knowledge, attitudes, and confidence of PI care in Taiwan. This study can be used as information for future app-based E-book education or related research designs for hard-to-heal wounds, such as diabetic ulceration and lower extremity ulceration, to reduce the comorbidities caused by wounds and the consumption of medical resources.

## Figures and Tables

**Figure 1 ijerph-19-15826-f001:**
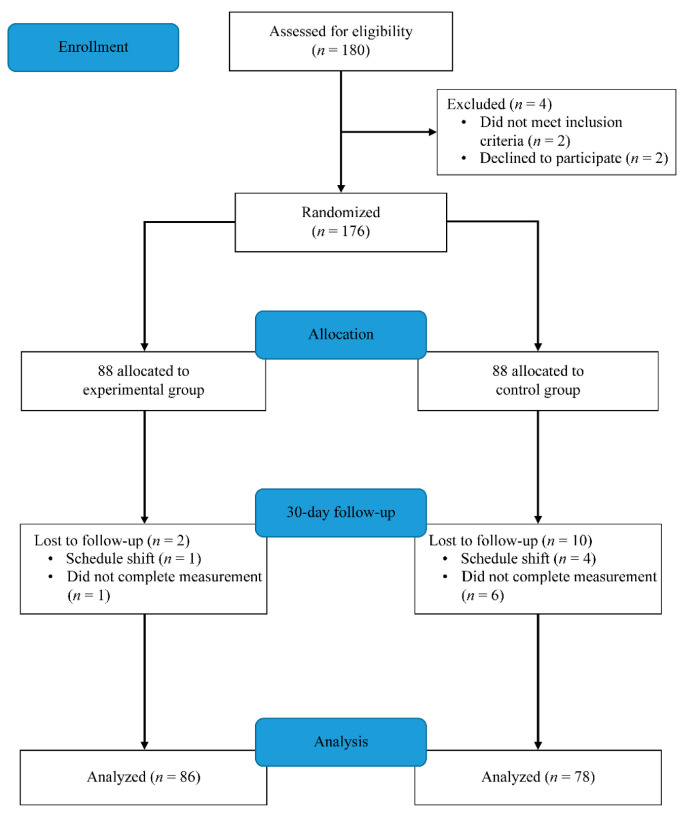
CONSORT flow diagram of the phases of the randomized trial.

**Table 1 ijerph-19-15826-t001:** Demographics of the study population (N = 164).

Variable	Total (*N* = 164)		Experimental Group (*n* = 86)		Control Group (*n* = 78)		χ^2^/t	*p*
	Mean	SD	Mean	SD	Mean	SD		
Age	31.28	6.29	32.35	6.21	30.15	6.20	0.143	0.706
Range	22.63–54.10		23.51–54.10		22.63–47.55			
Nursing work experience							0.001	0.975
years	6.03	5.52	7.18	5.89	4.94	4.94		
Range	0.5–28.17		0.5–28.17		0.5–20.00			
	n	%	n	%	n	%		
Marital status							5.25	0.154
Unmarried	120	73.2	61	70.9	59	75.6		
Married	34	20.7	21	24.4	13	16.7		
Others	10	6.1	4	4.7	6	7.7		
Education level							2.10	0.552
High school	4	2.5	2	2.3	2	2.6		
Junior college	52	31.7	26	30.2	26	33.3		
College	104	63.4	57	66.3	47	60.3		
Master or above	4	2.4	1	1.2	3	3.8		
Nursing level							2.98	0.560
Nurse trainees	21	12.8	10	11.6	11	14.1		
N0	25	15.2	15	17.4	10	12.8		
N1	29	17.7	13	15.1	16	20.5		
N2	71	43.3	36	41.9	35	44.9		
N3 or above	18	11.0	12	14.0	6	7.7		
Wound care course in school							0.90	0.343
Is not enough	36	21.9	28	32.6	8	10.3		
Insufficient	54	32.9	29	33.7	25	32.1		
Still can	68	41.5	27	31.4	41	52.6		
Enough	6	3.7	2	2.3	4	5.0		
In-service PI training							4.86	0.433
Is not enough	4	2.4	3	3.5	1	1.3		
Insufficient	38	23.2	24	27.9	14	17.9		
Still can	91	55.5	43	50.0	48	61.6		
Enough	31	18.9	16	18.6	15	19.2		
Very enough								
Received in-service PI education course							5.11	0.024
Yes	52	31.7	34	39.5	18	23.1		
No	112	68.3	52	60.5	60	76.9		
Reading PI articles							0.129	0.720
Yes	41	25.0	22	25.6	19	24.4		
No	123	75.0	64	74.4	59	75.6		

Note: Abbreviations: SD—standard deviation; PI—pressure injury.

**Table 2 ijerph-19-15826-t002:** Baseline score on KPIS, APIS, and CPIS in the experimental and control groups.

Variable	Total (*N* = 164)			Experimental (*n* = 86)			Control (*n* = 78)			*t*		*p*
	*M*	*SD*	Range	*M*	*SD*	Range	*M*	*SD*	Range			
KPIS	14.55	3.62	3–28	14.94	3.32	6–20	14.12	3.96	3–28	1.42	0.155	
APIS	84.85	12.37	68–120	83.88	10.41	68–113	84.11	9.04	68–120	−0.15		0.880
CPIS	106.57	20.84	33–153	103.68	18.36	67–153	105.74	16.53	67–144	−0.75		0.454

Abbreviations: *M* = mean; *SD* = standard deviation; KPIS = Knowledge of Pressure Injury Scale; APIS = Attitude of Pressure Injury Scale; CPIS = Confidence of PI Care Scale.

**Table 3 ijerph-19-15826-t003:** Comparison of knowledge, attitudes, and confidence scores between the two groups.

Variable	Experimental (*n* = 86)		Control (*n* = 78)		*t* ^b^ *(p)*		
	*M*	*SD*	*M*	*SD*			95%CI
KPIS							
Pretest	14.94	3.32	14.12	3.96	1.42 (*p* = 0.155)		−3.11 to 1.93
Posttest	24.16	3.99	16.42	3.17	13.65 (*p* < 0.000)		6.62 to 8.85
	*t*^a^ = −23.89 (*p* < 0.000)		*t*^a^ = −8.98 (*p =* 0.010)				
95% CI	−9.98 to −8.45		−2.80 to −1.78				
APIS							
Pretest	83.88	10.41	84.11	9.04	0.26 (*p* = 0.795)		−2.30 to 3.00
Posttest	106.63	10.68	92.57	10.30	8.57 (*p* < 0.0000)		10.82 to 17.30
	*t*^a^ = −24.73 (*p* < 0.000)		*t*^a^ = −10.24 (*p* < 0.000)				
95% CI	−23.95 to −20.39		−10.10 to −6.81				
CPIS							
Pretest	103.68	18.36	105.74	16.53	−0.63 (*p* = 0.529)		−6.86 to 3.54
Posttest	131.61	13.71	118.69	17.68	5.25 (*p* < 0.000)		8.06 to 17.77
	*t*^a^ = −20.51(*p* < 0.000)		*t*^a^ = −12.52 (*p* < 0.000)				
95% CI	−30.20 to −24.86		−15.00 to −10.89				

Note: *t* indicates the comparison of means between pretest and posttest scores within the group. ^a^. Paired-sample *t* test. ^b^. independent-sample *t* test.

## Data Availability

The data presented in this study are available on request from the corresponding author.

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
