# Peer review of "Effectiveness of an E-Book App on the Knowledge, Attitudes and Confidence of Nurses to Prevent and Care for Pressure Injury"

_ijerph, 2022, doi:10.3390/ijerph192315826_

Round 1
Reviewer 1 Report
Thank you for your work. This is an interesting study. I think readers would be interested to see the the PI e-book. Please can you cite a source for this material? Please can you comment in the discussion about the adequacy of PI knowledge in relation to the post-test KPIS scores in the 2 groups; effectively 67% vs 46%. Others scores, for example, suggest much higher % scores reflect adequacy. There should be some discussion about this. Given the data are very old - well before COVID, it is not really relevant to introduce COVID into your conclusions. Please comment about the age of your data in the implications. For example, could baseline PI knowledge be assumed to be the same today?
Author Response
Reviewer 1
- Please can you cite a source for this material?
Answer: Thanks to the reviewer for your reminders and suggestions. Regarding the material, we have explained in section 2.7; interventions parts (line 147-152). - Please can you comment in the discussion about the adequacy of PI knowledge in relation to the post-test KPIS scores in the 2 groups; effectively 67% vs 46%. Others scores, for example, suggest much higher % scores reflect adequacy.
Answer: Thanks to the reviewer for your reminders and suggestions. The explanation is as follows:
Like Western countries, most nursing schools in Taiwan only teach simple wound and drainage tube care in basic nursing experiential courses, which limits their knowledge of wound care. In this study, participants in both the experimental and control groups increased their knowledge after the systematic course (line 312-317). - Given the data are very old - well before COVID, it is not really relevant to introduce COVID into your conclusions.
Thanks to the reviewer for your reminders and suggestions. We have revised this explanation in the conclusion (please see 5. CONCLUSION, line 383-391).
Answer: The nursing shortage is a growing and urgent concern worldwide. How nursing staff can improve their PI knowledge and enhance the quality of care through effective learning methods without affecting their personal time is currently an important issue in clinical nursing. E-book apps allow them to self-regulate their learning, manage their time, seek assistance, and perform self-evaluations (An, Oh, & Park, 2022). Innovative education strategies offer a promising alternative for PI prevention and care for nurses. The E-book app has the potential to not only effectively improve nurses’ knowledge of PI and strengthen their confidence in wound management but improve their ability to perform safe and reliable professional wound care activities (Karimian et al., 2020). - Please comment about the age of your data in the implications. For example, could baseline PI knowledge be assumed to be the same today?
Thanks to the reviewer for your reminders and suggestions. An explanation is in the study limitation section (line 375-381).
Answer: This study was performed between 2014 to 2016, which are older research results. However, according to Martinengo et al (2020), who carried out a systematic review of 7 wound education studies for nursing students and clinical nurses, the results show that the intervention majority was online digital education; no interactive learning app design. Smartphones are popular and functionally enhanced, and future research can verify the effectiveness based on these study results.

Reviewer 2 Report
In the midst fourth industrial revolution era, research regarding the effect of innovative educational method is necessary to teach current nursing students and young nurses called as digital native. Here are some comments on this paper.
1. Please describe in detail how to conduct intervention in control and experimental group. How often and how long did authors conduct the intervention in each group?
2. Could you upload the figure of e-book ?
3. Please describe in detail who and how to teach the method to use of e-book.
4. Please describe in detail who conduct to measure outcomes. I would like to know whether the third person conduct to measure outcomes or not.
5. How did authors treat the missing data? Which method did you conduct in analysis, ‘intention to treat’ or ‘per protocol’?
6. This paper was measured the knowledge, attitude, and confidence, but was not measured a competency of caring PI. As identifying the effectiveness of e-book about care of PI, I think that a competency is one of the most important outcome. However, it is not easy to measure the competency in caring of PI. If nurses increase knowledge, attitude, or confidence of in caring of PI, competency would be increased. Authors should describe the relationship between competency and other variables which authors measured in this paper.
7. Why do authors think that the effectiveness of an innovation technology learning program in knowledge, attitudes, and confidence was better than in the conventional learning program? As authors described on page 7 “Although numerous studies on nurses’ knowledge of a PI training program revealed contradictory findings (Beeckman et al., 2011; Esche et al., 2015; Fulbrook et al., 2019)”, there are many debate. Therefore, authors should describe the advantage of innovation technology learning program including ‘repetition’, ’using high technology’, ‘at any time to help” and so on(An et al., 2022 : Jiwon An, Juyeon Oh, and Kyongok Park (2022). Self-Regulated Learning Strategies for Nursing Students: A Pilot Randomized Controlled Trial, Int. J. Environ. Res. Public Health 2022, 19, 9058. https://doi.org/10.3390/ijerph19159058
Author Response
Thank you for your comments. The response is in the word file. Please check it.

Round 2
Reviewer 2 Report
1. I think that it would be better to upload the figure as an appendix or supplementary file.
2. Please describe the initial of the author's name instead of the last name in line 123, 161.
3. Please added the new references in the reference list such as An, Oh, & Park, 2022.
Thank you.
Author Response
Response to Reviewer 2 Comments
Point 1: I think that it would be better to upload the figure as an appendix or supplementary file.
Response 1: Thanks to the reviewer for your reminders and suggestions. We have been upload the figure as an appendix (in line 481).
Point 2: Please describe the initial of the author's name instead of the last name in line 123, 161.
Response 2: Thanks to the reviewer for your reminders and suggestions. We have been
revised initial of the author's name in line114, 123, 161.
Point 3: please added the new references in the reference list such as An, Oh, & Park, 2022.
Response 3: Thanks to the reviewer for your reminders and suggestions. We have been added the new references in the reference list(line 407-408).